# Chemical-Reactivity Properties, Drug Likeness, and Bioactivity Scores of Seragamides A–F Anticancer Marine Peptides: Conceptual Density Functional Theory Viewpoint

**Norma Flores-Holguín** [1,†] **, Juan Frau** [2,†] **and Daniel Glossman-Mitnik** [1,2,*,†]

1    Laboratorio Virtual NANOCOSMOS, Departamento de Medio Ambiente y Energía, Centro de Investigación en Materiales Avanzados, Miguel de Cervantes 120, Complejo Industrial Chihuahua, Chihuahua, Chih 31136, Mexico; norma.flores@cimav.edu.mx
2    Departament de Química, Universitat de les Illes Balears, 07122 Palma de Mallorca, Spain; juan.frau@uib.es
*    Correspondence: daniel.glossman@cimav.edu.mx; Tel.: +52-6144391151
†    These authors contributed equally to this work.

**Abstract:** A methodology based on concepts that arose from Density Functional Theory (CDFT) was chosen for the calculation of global and local reactivity descriptors of the Seragamide family of marine anticancer peptides. Determination of active sites for the molecules was achieved by resorting to some descriptors within Molecular Electron Density Theory (MEDT) such as Fukui functions. The pKas of the six studied peptides were established using a proposed relationship between this property and calculated chemical hardness. The drug likenesses and bioactivity properties of the peptides considered in this study were obtained by resorting to a homology model by comparison with the bioactivity of related molecules in their interaction with different receptors. With the object of analyzing the concept of drug repurposing, a study of potential AGE-inhibition abilities of Seragamides peptides was pursued by comparison with well-known drugs that are already available as pharmaceuticals.

**Keywords:** Seragamides; chemical-reactivity theory; conceptual DFT; global and local reactivity descriptors; pKa; AGE-inhibition abilities; bioavailability; bioactivity scores

---

## 1. Introduction

There are numerous natural resources in the sea that are able to generate molecules that can act as a guide toward new medicine being created. It is due to this that most studies that have been undertaken in the recent past have had a direction towards new products where marine-species knowledge is used as important source of information [1]. One of the chemical species that stand out among those that can be obtained from natural products with marine origin are peptides, which are chemicals with a size between that of proteins and amino acids [2].

Peptides are being used in some therapeutic functions and they are referred to as therapeutic peptides. This is due to great possibilities with regards to aiding in the treatment of many diseases; consequently, there is much current research in this area [2]. From a medical viewpoint and considering the therapeutic peptides, knowledge of their bioactivity as well as their properties at the molecular level is of importance. The chemical reactivity and bioactivity of these peptides have a close relationship from a molecular viewpoint [3,4]. It is from this perspective that we conducted a study on chemical reactivity in natural peptides.

As a result, we deemed it crucial to study natural products' chemical reactivity, which is likely to help with the creation of some medicines by making use of tools presented through molecular modeling and computational chemistry. At present, in molecular modeling and computational chemistry, we have Conceptual Density Functional Theory (Conceptual DFT) [5–7] as the most powerful tool that is currently available for studying the chemical reactivity of molecular systems. Conceptual DFT, which is also known as Chemical Reactivity Theory, is capable of predicting the relationship of the way that chemical reactions take place by applying a series of global and local descriptors [8–10].

With the knowledge of chemical reactivity being essential for the development of new medicine, we investigated peptides obtained from marine sponges. The peptides were characterized and isolated, and the hope is that this could be a new therapeutic-peptide source [2,11]. The objective of this study was to investigate the level of chemical reactivity of Seragamides A–F anticancer marine peptides by use of Conceptual DFT techniques. It also involves the determination of local and the global properties, and, by so doing, it is possible for active reaction sites to be predicted and understood, in cases of both nucleophilic and electrophilic sites. On the basis of a methodology that we developed, the prediction of pKa values for each peptide is also possible [12]. We also outline the ability of therapeutic peptides acting as inhibitors attributable to Advanced Glycation Endproducts (AGEs) being formed using our previous ideas [13]. The calculation of bioactivity scores (descriptors of bioactivity) involving various methods is addressed in the literature [14,15]. With this approach, this study acts as a follow-up to previously published results on the family of marine-origin therapeutic peptides known as Mirabamides A–H [16].

## 2. Theoretical Background and Computational Methodology

Kohn–Sham (KS) theory involves the calculation of system energy, molecular density, and orbital energies, specifically associated with frontier orbitals including the Highest Occupied Molecular Orbital (HOMO) and Lowest Unoccupied Molecular Orbital (LUMO) [17–20]. This theory is handy when coming up with quantitative values associated with Conceptual DFT descriptors. At present, the application of range-separated (RS) exchange correlation functionals in Kohn–Sham DFT is of great concern [21–24]. There is a tendency for these particles to split the exchange and the $r_{12}^{-1}$ operator into long- and short-ranged parts; associated range-separation parameter $\omega$ that regulates the rate at which long-range behavior is acquired. The value of $\omega$ can either be fixed or "tuned" by use of a system-by-system mechanism where there is minimization of tuning standards. The optimal tuning style is based on having HOMO $\epsilon_H(N)$ energy for the case where KS theory applies. To have generalized KS theory applicable to an N electron system, we must have -IP (N), in which case IP represents the vertical ionization potential, which is an estimation of the energy difference, and E(N-1)-E(N), where we have the focus on a specific functional. For the case where we have the consideration of an approximate density functional, there maybe some significant difference between -IP (N) and $\epsilon_H$(N.) Optimal tuning entails establishing $\omega$, a system-specific range-separation parameter, by a non-empirical approach and having an RSE density functional [25–32]. Even with the absence-equivalent form that can be used for comparison with the prescription given for electron affinity (EA) coupled with LUMO energy in the case of presence neutral species, a conclusion can be that $\epsilon_H((N+1)= -EA(N)$ is possible. This makes the acquisition of the optimized $\omega$ value easier, which is that used in the prediction of both properties. Through this, Conceptual DFT descriptor prediction is enhanced. The concurrent prescription is dubbed as the "KID procedure" (for Koopmans in DFT), this being in reference to the relationship it has with Koopmans' theorem that has also been quoted in [16,33–39].

With reference to PubChem (https://pubchem.ncbi.nlm.nih.gov), this study was able to attain the molecular structures of the peptides Seragamides A–F. The website acts as public repository for information related to chemical substances and their related biological activities. In order for the resultant system to be preoptimized, we chose the most stable conformers. In making the choices, random sampling was used that involved molecular-mechanics methods with different torsional

angles involved. This was done through the overall MMFF94 force field [40–44] with connection to the MarvinView 17.15 program (ChemAxon, Budapest, Hungary). MarvinView is an advanced chemical viewer that is believed to be suitable for single and multiple chemical structures, reactions, and queries. This was followed by a review of the chemistry involved in the structures and a production of stereoisomer 3D structures, as well as by the use of MarvinView 17.15. As in the previous explanation, we further refined the subsequent geometries and then proceeded to a selection of the minimal energy conformation for every molecule for analysis of the HOMO and LUMO orbitals, as well as electronic energy at the DFT functional level.

In conformation with previous works [16,33–39], we performed computational studies by employing Gaussian 09 series of programs (Gaussian Inc., Wallingford, CT, USA), which are used in the implementation of density functional methods. We used the Def2SVP basis set with regard to geometry optimizations and in the determination of frequencies. In the calculation and analysis of electronic properties, the Def2TZVP basis set was applied [45,46]. In performing all calculations, water was chosen as the solvent in the Solvation Model Density (SMD) parameterization of the Integral Equation Formalism-Polarized Continuum Model (IEF-PCM) [47]. To establish the molecular structure and properties exhibited by the systems under study, the MN12SX density functional was selected due to it being known for producing results that are satisfactory in a number of structural and thermodynamic properties [48]. We considered the MN12SX/Def2TZVP/H20 model chemistry because previous research found that MN12SX behaves as a Koopmans-complaining density functional. This is useful to obtain HOMO and LUMO orbital energies that allow avoiding the calculation of cationic and anionic species whose convergence is sometimes difficult. However, we are exploring other density functionals to be considered for future works. We entered Simplified Molecular Input Line Entry Specification (SMILES) notations for compounds under study in the online Molinspiration software from Molinspiration Cheminformatics (Slovensky Grob, Slovak Republic). This was done to determine the various molecular properties related to drugability and for Bioactivity Score prediction for a variety of drug targets (GPCR ligands, kinase inhibitors, ion channel modulators, enzymes, and nuclear receptors) through a homology modeling procedure.

## 3. Results and Discussion

As indicated in Section 2, we obtained the molecular structures of the optimized conformers of the peptides Seragamide A–F which are displayed in Figure 1. This was achieved by considering the Density Functional Tight-Binding Approximation (DFTBA) model that is accessible in Gaussian 09. There was reoptimization in the gas phase; then, through the application of the MN12SX density functional with the Def2SVP basis set and the SMD solvent model, using water as the solvent, we had reoptimization again. Upon verification that there was matching in every one of the structures to the lowest energy conformations by employing the frequency-calculation analysis technique, we established the electronic properties of the same model chemistry. However, in this case, we had the Def2TZVP basis set as opposed to that used in the geometry optimization.

According to Becke, a common misconception exists in the connection between the KS electronic ground states and excitation energies [49]. Baerends et al. stated that the level of energy excitation within a KS system can be used as an effective measure for optimization of the molecular optical gap [50]. Thus, the HOMO–LUMO gap of the KS model was used to approximate excitation energy [51]. Ground-state calculations were used in the determination of the optimal maximum absorption wavelength that belongs to the fungal peptides of the Phallotoxin family based on the selected density functional to find the respective $\lambda_{max}$ values through the application of theoretical models that allow to establish the HOMO–LUMO gaps. Therefore, calculation of the maximum wavelength absorption of the Phallotoxin fungal peptides involved conducting ground-state calculations with the aforementioned density functional at the same level of model chemistry and theory, and determining the HOMO–LUMO gap, as can be seen in Table 1.

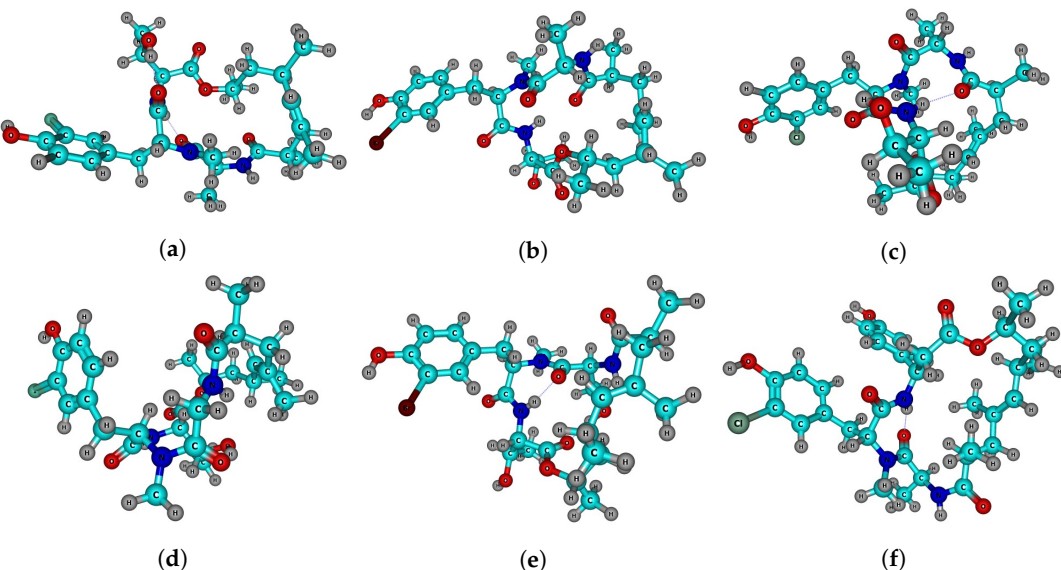

**Figure 1.** Graphical sketches of the molecular structures of: (**a**) Seragamide A; (**b**) Seragamide B; (**c**) Seragamide C; (**d**) Seragamide D; (**e**) Seragamide E; and (**f**) Seragamide F.

**Table 1.** HOMO and LUMO orbital energies as well as the HOMO–LUMO gap (in eV), and the maximum absorption wavelengths $\lambda_{max}$ (in nm) calculated with the MN12SX/Def2TZVP/$H_2O$ model chemistry.

| Molecule | HOMO | LUMO | HOMO-LUMO Gap | $\lambda_{max}$ |
|---|---|---|---|---|
| Seragamide A | −6.2902 | −0.9587 | 5.3315 | 233 |
| Seragamide B | −6.3789 | −0.7535 | 5.6254 | 220 |
| Seragamide C | −6.3247 | −0.7799 | 5.5449 | 224 |
| Seragamide D | −6.2567 | −1.3241 | 4.9326 | 251 |
| Seragamide E | −6.3422 | −1.0841 | 5.2581 | 236 |
| Seragamide F | −6.3038 | −0.8504 | 5.4534 | 227 |

For calculating global reactivity descriptors, as per past results for the melanoidin case [33–39], more so for peptides of marine origin [16], and with the MN12SX density functional being able to produce HOMO and LUMO where there a chance of verifying to what extent there is agreement with the results from Koopmans' theorem as an approximate methodology, the so-called KID procedure exists. Therefore, the use of the KID procedure is justified. Based on these considerations, the results in Table 1 are significant because they can be considered essential for the accurate calculation of the chemical reactivity and the $\lambda_{max}$ of each peptide from the consideration of the orbital energies and, as we present present, for the determination of their unknown pKa through consideration of a simple QSAR relationship previously developed by our research group. It is interesting to note that, while the HOMO value was almost the same for all the studied peptides, there were significative differences between their LUMOs, particularly for Seragamides D and E, a fact that is reflected on their chemical reactivities and pKas.

Through incorporation of the KID procedure that was considered in previous studies with regards to finite-difference approximation, the global reactivity descriptors may be given as [5,7,52,53]:

Electronegativity $\qquad \chi = -\frac{1}{2}(I + A) \approx \frac{1}{2}(\epsilon_L + \epsilon_H)$

Global Hardness $\qquad \eta = (I - A) \approx (\epsilon_L - \epsilon_H)$

Electrophilicity $\qquad \omega = \frac{\mu^2}{2\eta} = \frac{(I+A)^2}{4(I-A)} \approx \frac{(\epsilon_L+\epsilon_H)^2}{4(\epsilon_L-\epsilon_H)}$

Electrodonating power $\qquad \omega^- = \frac{(3I+A)^2}{16(I-A)} \approx \frac{(3\epsilon_H+\epsilon_L)^2}{16\eta}$

Electroaccepting power $\qquad \omega^+ = \frac{(I+3A)^2}{16(I-A)} \approx \frac{(\epsilon_H+3\epsilon_L)^2}{16\eta}$

Net electrophilicity $\qquad \Delta\omega^\pm = \omega^+ - (-\omega^-) = \omega^+ + \omega^-$

where $\epsilon_H$ and $\epsilon_L$ are the HOMO and LUMO energies associated with each of the peptides.

From the preceding discussion, the obtained results for the global reactivity descriptors on the basis of HOMO and LUMO energies with calculations based on the MN12SX density functional are outlined in Table 2.

**Table 2.** Global reactivity descriptors of Seragamides A–F calculated with the MN12SX/Def2TZVP/H$_2$O model chemistry.

| Molecule | Electronegativity $\chi$ | Global Hardness $\eta$ | Electrophilicity $\omega$ |
|---|---|---|---|
| Seragamide A | 3.6244 | 5.3315 | 1.2320 |
| Seragamide B | 3.5662 | 5.6254 | 1.1304 |
| Seragamide C | 3.5523 | 5.5449 | 1.1379 |
| Seragamide D | 3.7904 | 4.9326 | 1.4564 |
| Seragamide E | 3.7131 | 5.2581 | 1.3111 |
| Seragamide F | 3.5771 | 5.4534 | 1.1732 |
| **Molecule** | **Electrodonating Power $\omega^-$** | **Electroaccepting Power $\omega^+$** | **Net Electrophilicity $\Delta\omega^\pm$** |
| Seragamide A | 4.6093 | 0.9849 | 5.5943 |
| Seragamide B | 4.3954 | 0.8293 | 5.2247 |
| Seragamide C | 4.3985 | 0.8462 | 5.2447 |
| Seragamide D | 5.1162 | 1.3528 | 6.4420 |
| Seragamide E | 4.8073 | 1.0942 | 5.9015 |
| Seragamide F | 4.4757 | 0.8986 | 5.3743 |

As per the expectations from the molecular structure exhibited by this species, its electrodonating nature outweighs its electroaccepting nature. As the methodology employed for the calculation of the global reactivity descriptors relies on the accurate determination of HOMO and LUMO orbital energies, as mentioned above, it can be seen in Table 2 that, while there were no great differences between electronegativity $\chi$ for any peptides, the values were different for the case of global hardness $\eta$ and electrophilicity $\omega$. Global hardness represents a measure of electron-density resistance to deformation. This implies that the harder a molecule is, the less reactive it is. It can be concluded that Seragamide D was the most reactive peptide, while Seragamide B was the least reactive of the six molecules. This can also be seen with electrophilicity index $\omega$: it attained the maximal value for Seragamide D and the minimal for Seragamide B. Besides the indicated electrodonating- and electroaccepting-power behavior, net electrophilicity, which resulted from a comparison of both descriptors, also reflects these differences in chemical reactivity, being the maximum for Seragamide D and the minimum for Seragamide B.

A previous discussion focused on the application of conceptual DFT descriptors to evaluate the computation prediction of pKa peptides, where it was established that the pKa = 16.3088–0.8268 $\eta$ relationship would play an important role in the initial prediction of complex peptides, which are important in the manufacture of medical drugs [12]. Given the biological pH level, the peptides under study existed as neutral molecules and were still considered to be neutral during pKa computations [12]. The pKa relationship is also important in the optimization of the molecular structure of every conformer as well as the computation of pKa values for all molecules given the $\eta$ values shown in Table 2. The computational results of the pKa values for the Seragamide molecules are shown in Table 3.

**Table 3.** pKas of Seragamides A–F.

| Molecule | pKa |
|---|---|
| Seragamide A | 11.90 |
| Seragamide B | 11.66 |
| Seragamide C | 11.72 |
| Seragamide D | 12.23 |
| Seragamide E | 11.96 |
| Seragamide F | 11.80 |

The pKa values shown in Table 3 indicate that the used computational methodology was effective in the differentiation of the respective pKa values for all peptide molecules irrespective of the significance of the difference. It can be seen that the pKa for Seragamide D presented the maximal value, which resulted from the employed methodology relying on global hardness and its negative sign in QSAR expression. The pKa values of these peptides are important in the manufacture of pharmaceutical drugs by explaining procedures used in drug delivery and their respective action mechanisms.

The Maillard reaction observed between amino group peptides or proteins and a reducing carbonyl results in a Schiff base being formed. Through a series of steps, it renders AGE molecules. It is believed that, when AGEs exist, they are one of the main reasons that result in the development of some diseases such Parkinson's disease, diabetes, and Alzheimer's disease [54]. At the backdrop of the many strategies that have been used in managing and controlling AGE formation, it may be important to remark that the use of amino groups presenting compounds in structure as being competitive considering amino acids, proteins, and peptides that are found in the body and that have the capability of interacting with carbohydrate carbonyl, which has a reducing ability. Some compounds have been drug-formulated to be able to fulfill this objective. Some typical examples are Tenilsetam, Metformin, Pyridoxamine, Pioglitazone, Carnosine, and Aminoguanidine [55,56]. One can say that peptides with amino and amido groups may serve as therapeutic drugs that hinder the formation of any AGEs. This is because there can be a Maillard reaction involving the reduction of carbohydrates ahead of body peptides and proteins. Even though this may just be a hypothesis, it is worth exploring the chance by following the above methodology in this paper. In studies prior to this one, we investigated a group of some suggested molecules to act as inhibitors in AGE formation through observation of their behavior in connection with Conceptual DFT reactivity descriptors [13]. Therefore, it was concluded that chemical reactivity for potential AGE inhibitors relying on their nucleophilic character was the main factor, regardless of definitions given to nucleophilicity. The findings in the study indicated that net electrophilicity inverse $\Delta\omega^{\pm}$ could serve well as a nucleophilicity N definition. As per the discussed analysis, some qualitative trends were identified in the molecular systems under investigation. In this study, we extended the correlation to Seragamides A–F so that the possibility could be established of them as precursors of therapeutic drugs that can inhibit AGE formation, in addition to their known function as anticancer peptides. As the chemistry model employed in both studies was the same, the comparison was straightforward:

$$\text{ALT-946} > \text{Aminoguanidine} > \text{Metformin} > \text{Carnosine} > \text{Seragamide B} >$$
$$> \text{Seragamide C} > \text{Seragamide F} > \text{Tenilsetam} > \text{Seragamide A} >$$
$$> \text{Pyridoxamine} > \text{Seragamide D} > \text{Pioglitazone} > \text{Seragamide E}$$

The qualitative behavior can be seen as a representation of recognized pharmacological properties of AGE inhibitors that were studied, It is clear that peptides under study have AGE inhibitor abilities that are also associated with Pyridoxamine, with some having better performance but others registering lower values of AGE inhibition.

Within Conceptual DFT, the Fukui function indicates changes in the electron density of a molecule at a given position when the number of electrons have changed [5,7]. Thus, the Fukui function allows predicting where the most electrophilic and nucleophilic sites of a molecule are. The Fukui function has two finite versions of the associated change that depend on whether or not an electron was removed or added from the molecule. Applying the same ideas as before, definitions for these local reactivity descriptors are [5–7]:

$$\text{Nucleophilic Fukui Function} \qquad f^{+}(\mathbf{r}) = \rho_{N+1}(\mathbf{r}) - \rho_{N}(\mathbf{r})$$

$$\text{Electrophilic Fukui Function} \qquad f^{-}(\mathbf{r}) = \rho_{N}(\mathbf{r}) - \rho_{N-1}(\mathbf{r}),$$

which are the relationship between the electronic densities of neutral, positive, and negative species. In turn, the dual descriptor $\Delta f(\mathbf{r})$ [57–62] is defined as:

$$\text{Dual Descriptor} \qquad \Delta f(\mathbf{r}) = f^{+}(\mathbf{r}) - f^{-}(\mathbf{r})$$

A graphical representation of dual descriptor $\Delta f(\mathbf{r})$ for Seragamide A–F molecules is shown in Figure 2.

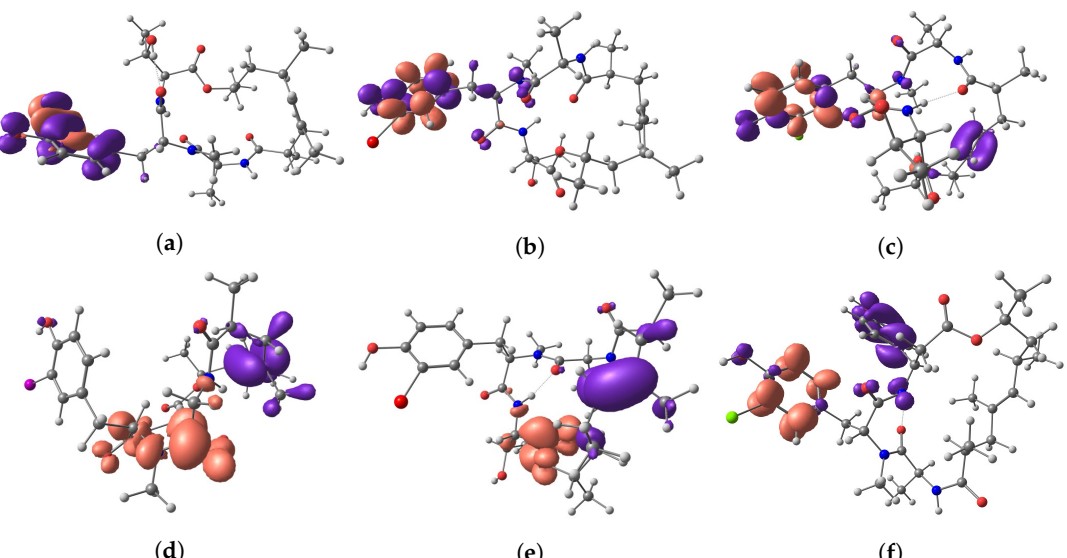

**Figure 2.** Graphical sketches of the Dual Descriptor $\Delta f(\mathbf{r})$ of: (**a**) Seragamide A; (**b**) Seragamide B; (**c**) Seragamide C; (**d**) Seragamide D; (**e**) Seragamide E; and (**f**) Seragamide F.

The lobe positions in each case indicate where the electrophilic and nucleophilic regions within the peptides are located, the darker ones being those belonging to $\Delta f(\mathbf{r}) < 0$ and the clearer ones corresponding to $\Delta f(\mathbf{r}) > 0$. This depends on the molecular structure and the electronic density that results from it. For the studied peptides, these regions were mainly located around the phenyl rings. It can be seen in Figure 2 that different substituents exerted important influence for the I, Cl, and Br atoms in Seragamides A–C, respectively, the addition of a second phenyl ring in Seragamide F, or a $CH_2OH$ group in Seragamide E.

The degree of oral bioavailability of molecules or drugability that can potentially be used in the manufacture of drugs was measured using the Lipinski Rule of Five by determining molecules that possessed drug-like properties, as shown in Table 4.

**Table 4.** Molecular properties of Seragamides A–F calculated to verify the Lipinski Rule of Five.

| Molecule | LogP | TPSA | nAtoms | nON | NOHNH |
|---|---|---|---|---|---|
| Seragamide A | 3.69 | 145.27 | 40 | 10 | 4 |
| Seragamide B | 3.42 | 145.27 | 40 | 10 | 4 |
| Seragamide C | 3.29 | 145.27 | 40 | 10 | 4 |
| Seragamide D | 3.36 | 145.27 | 39 | 10 | 4 |
| Seragamide E | 2.44 | 165.49 | 41 | 11 | 5 |
| Seragamide F | 3.73 | 145.27 | 45 | 10 | 4 |

| Molecule | Nviol | Nrotb | Volume | MW | |
|---|---|---|---|---|---|
| Seragamide A | 1 | 3 | 547.28 | 671.57 | |
| Seragamide B | 1 | 3 | 541.17 | 624.57 | |
| Seragamide C | 1 | 3 | 536.82 | 580.12 | |
| Seragamide D | 1 | 3 | 530.69 | 657.55 | |
| Seragamide E | 2 | 4 | 549.43 | 640.57 | |
| Seragamide F | 1 | 3 | 591.64 | 642.19 | |

The results presented in Table 4 can be better understood if a graphical representation of these physicochemical properties is displayed in the form of what is known as a bioavailability radar, where six physicochemical properties are taken into account: lipophilicity, size, polarity, solubility, flexibility, and saturation. A physicochemical range for each descriptor is depicted as a pink area in which the radar plot of the molecule has to fall entirely to be considered drug-like. These results are presented in Figure 3.

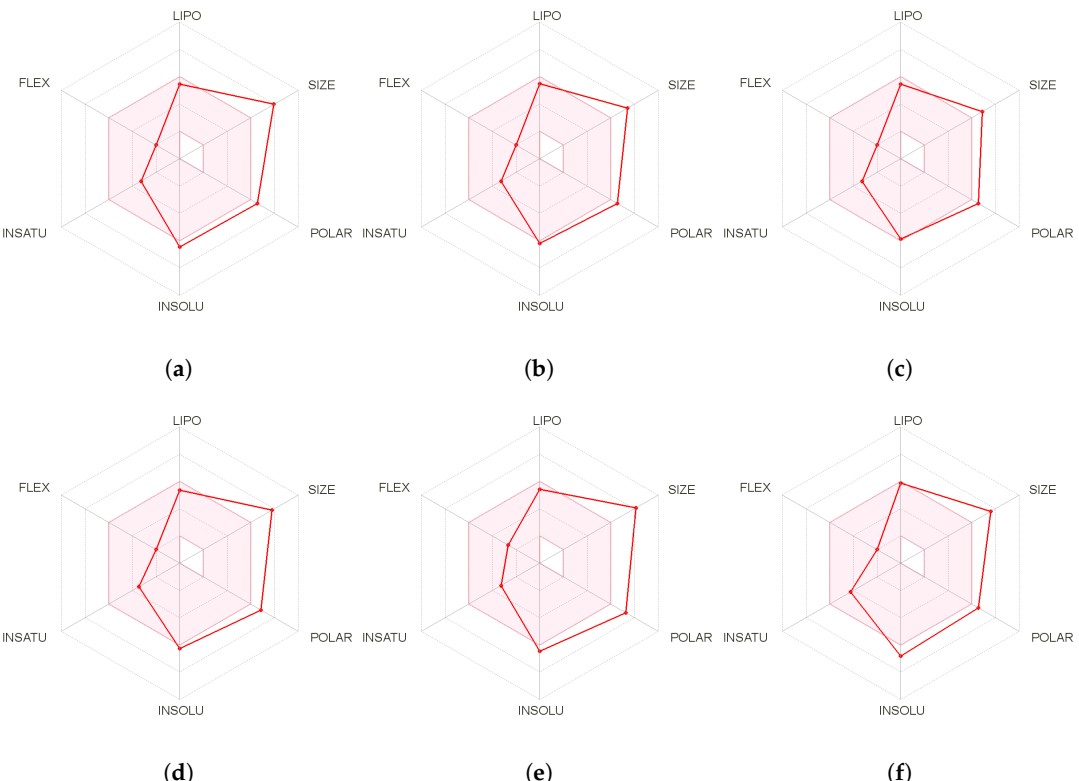

**Figure 3.** Bioavailability Radars of: (**a**) Seragamide A; (**b**) Seragamide B; (**c**) Seragamide C; (**d**) Seragamide D; (**e**) Seragamide E; and (**f**) Seragamide F.

It can be concluded from the analysis shown in Table 4 and the bioavailability radars in Figure 3 that the major drawback for considering peptides as potential therapeutic drugs is their large size, which is reflected in the associated molecular weight (MW) and, for the case of Seragamide E, also due to the high number of hydrogen bonds.

However, it has been reported in the literature that this technique cannot be applied in measuring the bioavailability of peptides due to the existence of hydrogen bonds and high molecular weight properties [63,64]. Thus, in this study, we applied a different technique in the evaluation of the chemical structure of other compounds that were predicted to possess similar bioactivity properties as the Phallotoxin peptides under study. As illustrated in Section 2, evaluation of the pharmacological properties of different compounds in the process of determining bioactivity properties can be carried out using Molinspiration software based on the variability of drug targets, as shown in Table 5.

**Table 5.** Bioactivity Scores of Seragamides A–F calculated on the basis of GPCR ligand, ion channel modulator, nuclear receptor kigand, kinase inhibitor, protease inhibitor and enzyme inhibitor interactions.

| Molecule | GPCR Ligand | Ion Channel Modulator | Kinase Inhibitor |
|---|---|---|---|
| Seragamide A | 0.23 | −0.22 | −0.40 |
| Seragamide B | 0.17 | −0.36 | −0.39 |
| Seragamide C | 0.26 | −0.27 | −0.31 |
| Seragamide D | 0.22 | −0.20 | −0.43 |
| Seragamide E | 0.20 | −0.39 | −0.39 |
| Seragamide F | 0.04 | −0.73 | −0.66 |
| **Molecule** | **Nuclear Receptor Ligand** | **Protease Inhibitor** | **Enzyme Inhibitor** |
| Seragamide A | 0.01 | 0.30 | 0.19 |
| Seragamide B | −0.09 | 0.27 | 0.18 |
| Seragamide C | −0.04 | 0.32 | 0.21 |
| Seragamide D | 0.03 | 0.28 | 0.22 |
| Seragamide E | −0.06 | 0.31 | 0.21 |
| Seragamide F | −0.50 | 0.14 | −0.24 |

According to the table, peptides whose bioactivity score was less than zero were considered to be active, while organic molecules whose bioactivity score was between zero and −5 were considered to be moderately active, and organic molecules with a score of less than −5 were considered to be inactive. All peptides that were considered during this study, with the exception of Seragamide F, were found to have moderate bioactivity scores to interact as a G-Protein Coupled Receptor (GPCR) ligand, which is an interesting property because it is estimated that approximately 35% of already approved drugs in the market target GPCRs. Moreover, it can be seen that all peptides were moderately bioactive to act as protease inhibitors and, with the exception of Seragamide F, also as enzyme inhibitors.

During the development process of a new drug, it is very important to know the possible fate of a therapeutic compound in the organism, a process that is known as pharmacokinetics. This can be done by analyzing the associated effects in the form of individual indices that are called Absorption, Distribution, Metabolism, and Excretion (ADME) parameters. These parameters can be obtained by using computer models that can be an alternative to the experimental procedures for their determination.

In this work, some ADME parameters were estimated with the aid of the online available SwissADME software [65], and the results are presented in Table 6.

**Table 6.** ADME parameters related to pharmacokinetics of Seragamides A–F.

| ADME | Seragamide A | Seragamide B | Seragamide C | Seragamide D | Seragamide E | Seragamide F |
|---|---|---|---|---|---|---|
| GI absorption | Low | Low | Low | Low | Low | Low |
| BBB permeant | No | No | No | No | No | No |
| P-gp substrate | Yes | Yes | Yes | Yes | Yes | Yes |
| CYP1A2 inhibitor | No | No | No | No | No | No |
| CYP2C19 inhibitor | No | No | No | No | No | No |
| CYP2C9 inhibitor | No | No | No | No | No | No |
| CYP2D6 inhibitor | No | No | No | No | No | No |
| CYP3A4 inhibitor | Yes | Yes | Yes | Yes | Yes | Yes |
| Log $K_p$ (skin permeation) | $-7.52$ cm/s | $-7.21$ cm/s | $-6.98$ cm/s | $-7.56$ cm/s | $-7.65$ cm/s | $-6.83$ cm/s |

Predictions for passive human gastrointestinal (GI) absorption and blood–brain barrier (BBB) permeation were the same for the six peptides: low for the first ADME parameter and negative for the second. Knowledge about compounds being substrate or non-substrate of the permeability glycoprotein (P-gp) is of utmost importance because its major role is to protect the central nervous system (CNS) from xenobiotics. It can be seen that all peptides considered in this work could be predicted as substrates of this glycoprotein. Knowledge of the interaction of potential therapeutic drugs with cytochromes P450 (CYP) is essential because this group of isoenzymes is a key player in drug elimination through metabolic biotransformation. The estimated results indicated that, for the six Seragamides, there would be no interaction with CYP1A2, CYP2C19, CYP2C9, and CYP2D6, but all would act as inhibitors of the CYP3A4 member of the isoenzyme family.

## 4. Conclusions

This paper presents findings regarding a chemical-reactivity study of a group of six peptides of marine origin, Seragamides A–F, that have the potential of exhibiting therapeutic properties based on Conceptual DFT as a tool to explain the molecular interactions. It is worth mentioning that the used tools did not register any significant errors, even with the relatively low computational efforts that were required, which was an advantage. The new findings on the values of global and local descriptors for the molecular reactivity for the peptides that were under study could be suitable in coming up with new drugs that are dependent on these compounds. Similarly, we predicted pKa values for each of the therapeutic peptides through chemical-hardness values that followed a previously suggested methodology, and the obtained data could be useful in gaining insight into chemical reactivity and other important properties, including water solubility. Finally, the results of homology modeling shed light on the predicted bioactivity and bioavailability of the considered peptides, followed by prediction of ADME parameters that could be helpful in the development of new therapeutic drugs based on these marine peptides.

**Author Contributions:** D.G.-M. conceived and designed the research and headed, wrote, and revised the manuscript; and N.F.-H. and J.F. contributed to the analysis of the results and the writing and revision of the article.

**Funding:** Consejo Nacional de Ciencia y Tecnología (CONACYT, Mexico) through Grant 219566-2014 and Ministerio de Economía y Competitividad (MINECO) and the European Fund for Regional Development through Grant CTQ2014-55835-R were the financial supporters of this study.

**Acknowledgments:** Daniel Glossman-Mitnik conducted this work as a Visiting Lecturer at the University of the Balearic Islands, from which support is gratefully acknowledged.

**Conflicts of Interest:** The authors declare no conflict of interest.

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
