# Peer review of "Chemical-Reactivity Properties, Drug Likeness, and Bioactivity Scores of Seragamides A–F Anticancer Marine Peptides: Conceptual Density Functional Theory Viewpoint"

_computation, doi:10.3390/computation7030052_

Round 1
Reviewer 2 Report
The manuscript has certain interesting but I feel that the authors can explain and discuss more the data obtained. In general, Tables are not properly discussed, i.e. huge amount of numbers but no explanation about the meaning.
I would suggest the authors to revise and correct the manuscript. My comments below:
Line 33: "DFT which is also known as Chemical Reactivity Theory" I guess the authors mean Conceptual DFT rather than DFT.
Line 37-46 It's not clear what authors mean with this paragraph. The methodology is ready to use and there are several benchmarks and "usage" on a wide variety of systems. If the authors want to use a certain methodology, they need to justify it and compare with data in the literature. From my humble opinion, this paragraph is to be removed.
Table 1. Total electronic energies are meaningless. If the authors want to present those values I would suggest to move them into the ESI and make a comment in the main text.
Also, the data in Table 1 is not discussed in the main text.
Greek symbols for the descriptors should be also included in Table 2 so the reader can follow the discussion. Also, 6 different descriptors are presented but the data is not discussed but only the relationship between global hardness and pKas. Besides, other molecule/values should be used to estimate the relative "power" of those obtained by the authors, i.e. values are needed to compare with, a standard to refer to.
How the pKas were obtained? The rest of the descriptors and methodology is quite detailed so one finds missing that last piece of information.
Line 166-191. That paragraph belongs to the introduction rather than to results.
Could the authors give (in the text) a chemical idea about what are the fukui functions? What are the colours stand for? How the Figure 2 should be interpreted? What is the relationship between lobes and reactivity? Is there any relationship between the volume of each lobe and the reactivity?
Again, Table 4 and 5. Lots of data but no explanation, discussion. In my opinion the rule of thumb is: If the data is not discussed, it should be moved into the ESI. All data in Tables should be properly discussed and integrated within the text.
Round 2
Reviewer 2 Report
The authors have adressed all my points and concerns. I see no further objection for publish the article as it stands.
